# The SMC Hinge is a Selective Gate for Obstacle Bypass

Hon Wing Liu [1,4], Florian Roisné-Hamelin [1,4], Michael Taschner [1,4], James Collier[2,3], Madhusudhan Srinivasan[2] & Stephan Gruber [1] ✉

DNA loop-extruding SMC complexes play vital roles in genome maintenance and DNA immunity. However, how these ring-shaped DNA motors navigate large DNA-bound obstacles has remained unclear. Here, we demonstrate that a bacterial SMC Wadjet complex can efficiently bypass obstacles larger than the SMC coiled coil lumen when they are tethered to the extruded DNA by a single-stranded DNA or RNA linker. This bypass is mediated by the selective entrapment of the linker within the SMC hinge channel, which functions as an obligate gate—permitting passage of the linker while retaining double-stranded DNA stably inside the SMC ring and keeping associated obstacles outside. We further show that eukaryotic SMC hinges similarly accommodate ssDNA, and that the hinge is dispensable for loop extrusion by Wadjet, altogether suggesting that obstacle bypass represents the conserved, long-sought function of the SMC hinge toroid. By integrating hinge bypass with loop extrusion, we provide a mechanistic framework for how DNA-entrapping SMC complexes can generate chromosomal loops densely decorated with obstacles.

Chromosomes fold into compact yet dynamic 3D structures through DNA loop extrusion (LE), driven by conserved ATP-powered structural maintenance of chromosomes (SMC) motors. LE is the enzymatic activity that underpins diverse genome organizational features across life, from bacterial chromosome arm alignment to eukaryotic interphase chromosome domain formation and mitotic chromosome condensation. LE brings distant DNA segments into proximity to regulate transcription, repair, and recombination[1–3]. SMC proteins harbor an extended coiled coil, with an ABC ATPase head domain at one end and a hinge domain at the other. They dimerize by forming a toroid-shaped hinge and bind a kleisin subunit at the heads, generating an elongated, ring-like structure[4–6]. The complex is further elaborated by either a KITE dimer (in bacterial SMC complexes and in Smc5/6) or two HAWK subunits (in cohesin and condensin) that bind to the kleisin subunit[7,8]. Purified SMC complexes can entrap and extrude DNA in vitro[9–15]. While several models for the mechanism of LE have been put forward[16–21], the exact molecular processes of LE remain unclear.

Wadjet, a family of SMC protein complexes, are bacterial immune system components that use LE to selectively recognize and restrict invading plasmids by DNA cleavage[22–25] while sparing the larger chromosomal DNA (Supplementary Fig. 1A). Our previous work suggested that DNA is topologically entrapped inside Wadjet SMC rings during LE[25]. If SMC-mediated LE serves as a universal mechanism for genome organization, it implies that the motors driving LE must navigate various obstacles on DNA, including nucleosomes, DNA replication and transcription machineries. In prokaryotes, ribosomes associated with transcription-translation complexes would also pose significant barriers to LE, further challenging the process. Although LE of "chromatinized" DNA has not been demonstrated in vitro, SMC motors have been observed to bypass individual nucleosomes. Strikingly, single-molecule studies show they can bypass other SMC complexes, forming interlocking "Z-loops," and can even overcome artificial obstacles much bigger than themselves[11,26,27]. Interestingly, the apparently efficient incorporation of large obstacles into DNA loops formed by

[1]Department of Fundamental Microbiology (DMF), Faculty of Biology and Medicine (FBM), University of Lausanne (UNIL), Lausanne, Switzerland. [2]Department of Biochemistry, University of Oxford, Oxford, UK. [3]MRC Laboratory of Molecular Biology, Cambridge, UK. [4]These authors contributed equally: Hon Wing Liu, Florian Roisné-Hamelin, Michael Taschner. ✉e-mail: stephan.gruber@unil.ch

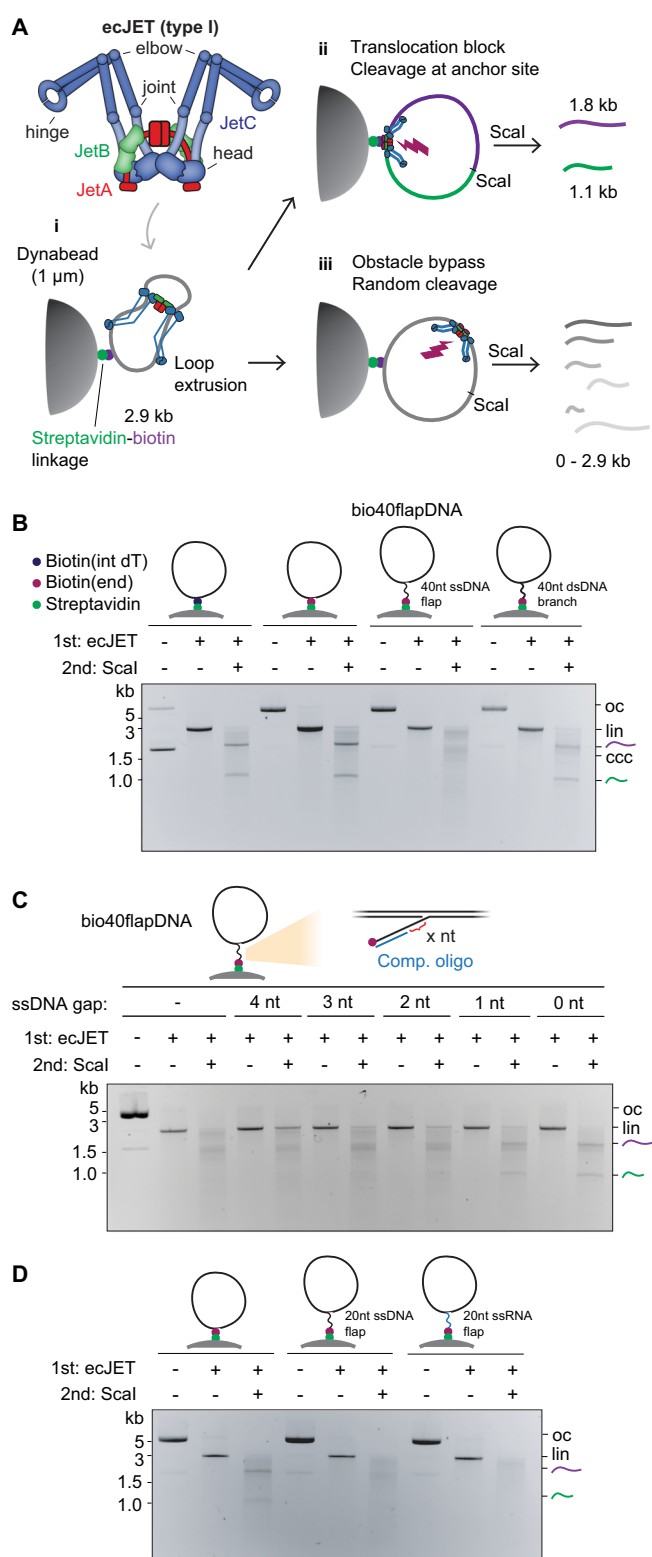

**Fig. 1 | Requirements for obstacle bypassing by Wadjet-I. A** Schematic depiction of Wadjet-I from *E. coli* strain GF4-3 (ecJET). SMC JetC (blue), KITE JetB (green), kleisin JetA (red). The JetC hinge domain and elbow are highlighted. i) Depiction of the extrusion-cleavage activity of Wadjet, on 2.9 kb DNA circles containing a Dynabeads obstacle (attached via biotin-streptavidin linkage). Two possible outcomes are depicted: ii) Wadjet stalls at the obstacle, triggering cleavage at the roadblock anchor point. Post-treatment by ScaI single cutter results in two defined fragments of 1.8 kb (purple) and 1.1 kb (green) in size. iii) One motor unit of Wadjet bypassing the obstacle, resulting in cleavage at a random position. Post-treatment by ScaI results in fragments of variable size. See Supplementary Fig. 1A for Wadjet activity on naked DNA. **B** Wadjet bypasses a large obstacle (Dynabeads) attached onto DNA by a ssDNA linker. Agarose gel showing the cleavage/bypass activity of ecJET on DNA circles containing various linkages to biotin. Note that untreated DNA circles containing 5'-positioned biotins (end) migrate slower at ~7 kb, as they contain a DNA nick and thus represent open circles (oc), circles with internally positioned biotin (int) are mostly covalently closed (ccc) and migrate fast in the presence of ethidium bromide. Linearized DNA (lin). See Supplementary Fig. 1B for details on DNA substrate generation. **C** Agarose gel showing the cleavage/bypass activity of ecJET on bio40flapDNA attached onto Dynabeads with partially complementary oligos (in blue) annealing at the ssDNA flap, leaving behind a ssDNA linker of defined lengths. **D** Agarose gel showing the cleavage/bypass activity of ecJET on DNA with an RNA-linked obstacle. See Supplementary Fig. 5 for further experiments on RNA-linked obstacles. Representative gel images from three technical replicates are shown. Schematics for panels **B**–**D** adapted from[25].

allowing passage of ssDNA and RNA but excluding dsDNA. We find that LE and obstacle bypass activities by SMC complexes are independent of each other and are contributed by different parts of the SMC ring.

## Results

### A single-stranded nucleic acid linker facilitates efficient obstacle bypass by Wadjet-I

Wadjet (JetABCD) SMC complexes restrict plasmids through DNA cleavage in diverse bacteria and archaea. They exist in three types (I, II, III, denoted here as Wadjet-I, -II, and -III) with differences in operon organization and domain composition[24,28]. Cleavage is DNA sequence non-specific and involves a dimeric JetABC motor actively extruding the plasmid, followed by activation of the JetD nuclease subunit once extrusion is complete (Supplementary Fig. 1A)[13,22,23,25]. We exploit this extrusion-cleavage activity to study obstacle bypassing, with DNA molecules cleaved at the obstacle position marking extrusion-cleavage events that occurred without bypassing (Fig. 1A). Wadjet-I (from *Escherichia coli* strain GF4-3, 'ecJET') cleaved DNA circles anchored to micrometer-scale Dynabeads obstacles via streptavidin-biotin linkage mostly at the obstacle anchor and only occasionally away from the anchoring point[25]. This observation suggests that ecJET can bypass bead obstacles, albeit inefficiently, raising the question of whether its intrinsic bypassing ability was limited by constraints imposed by the synthetic DNA-obstacle linkage itself[25,29].

To investigate this further, we systematically modified the positioning of the biotin moiety on the 2.9 kb DNA circle (Fig. 1B, Supplementary Fig. 1B). ecJET tended to cleave at the Dynabeads obstacle attachment point on DNA circles containing biotin positioned at the 5' end of a nick, as previously seen with internal biotin[25]. Strikingly, the cleavage site distribution became random if forty ssDNA nucleotides were placed between the plasmid DNA and the 5' end biotin (hereafter abbreviated as bio40flapDNA) (Supplementary Fig. 1B), indicating that the Dynabeads surprisingly posed little or no barrier for ecJET if attached via ssDNA. Given that extrusion of this short DNA substrate is expected to occur within seconds, this finding implies that bypassing is not only efficient but also fast. Notably, bio40flapDNA circles were as stably anchored on Dynabeads as in the other scenarios (Supplementary Fig. 2A), making it unlikely that the random cleavage pattern was due to DNA dissociation from the beads. Remarkably, converting the

permanently circularized SMC rings[11,27] led to the proposal that DNA loops might be held outside the SMC ring during LE[19]. Hence, the need for and existence of bypassing remain debated, its mechanisms are unknown, and the specific components of SMC complexes responsible have yet to be identified.

Here, we demonstrate efficient bypassing of large obstacles by an SMC complex and show that this process depends on access to single-stranded nucleic acid by the SMC hinge, which is a conserved feature among diverse SMCs. We elucidate the SMC hinge as a selective gate,

ssDNA flap to a dsDNA branch by annealing a complementary oligonucleotide reverted the cleavage pattern by ecJET to an anchor-site-specific one (Fig. 1B). This suggests that ecJET can indeed efficiently bypass DNA obstacles larger than its own dimensions—as long as the linkage to the DNA includes a certain extent of single-stranded DNA. By annealing shorter oligonucleotides (thus resulting in dsDNA-ssDNA hybrid flaps, Supplementary Fig. 1B), we found that a stretch of ten ssDNA nucleotides at the biotin end was sufficient to create a permeable barrier. Intriguingly, even two ssDNA nucleotides at the plasmid end enabled efficient bypass (Fig. 1C, Supplementary Fig. 2B). This suggests that bypass can occur both immediately adjacent to the plasmid DNA and at a distance from it, with the former configuration being more favorable for SMC bypass in our assay. Importantly, ssDNA stretches that were not part of the linker but were positioned adjacent to the bead anchor, such as a 10-nucleotide 3′ flap or a 12-nucleotide bubble near the biotin moiety, did not facilitate efficient bypass (Supplementary Fig. 2C). This highlights the critical role of ssDNA specifically within the linker region, rather than its mere presence near the anchor, in enabling ecJET to navigate around obstacles. We further asked whether obstacle linkers comprised of ssRNA, featuring similar chemistry, size, and charge distribution to ssDNA, also permitted bypass. Indeed, we observed efficient bypassing by ecJET of Dynabeads attached through a 20 nt RNA linker (Fig. 1D), indicating that the obstacle linker need not be of a specific chemical structure to be bypassed by ecJET, as discussed further below.

## The Wadjet-I hinge promotes obstacle bypass

SMC proteins typically adopt a donut-shaped hinge structure featuring a central channel lined with positively charged residues (Fig. 2A, Supplementary Fig. 3). Despite the high conservation of these charged residues across species, their specific functional role remains poorly understood. Intriguingly, this feature is absent from some SMC proteins including a Wadjet-II (from *Neobacillus vireti* strain LMG 21834, 'nvJET') (Fig. 2B, Supplementary Fig. 3), which, like ecJET, is capable of cleaving naked circular DNA[30]. Strikingly however, nvJET was totally bypass-incompetent, even with bio40flapDNA (Fig. 2B), resulting in DNA cleavage exclusively at the obstacle anchor position. This raises the interesting possibility that only hinges with a donut shape are capable of facilitating obstacle bypass by temporarily accommodating ssDNA/RNA in their channel and acting as single-strand-specific gates. To determine whether hinge domains must detach from one another for obstacle bypass, we utilized a Cys-less ecJET (that lacks all endogenous cysteine residues[31]) and introduced cysteine pairs at ecJetC hinge interfaces. Specifically, we engineered two derivatives containing either K529C/R593C or Q508C/H576C, enabling closure of the hinge donut via chemical crosslinking with 1,4-butanediyl bismethanethiosulfonate (M4M) (Fig. 2A, Supplementary Fig. 4A)[32]. Cys-less ecJET (WT) was proficient in plasmid cleavage and efficiently bypassed Dynabeads on bio40flapDNA (Fig. 2C). While hinge crosslinking in both cysteine pair variants did not hinder DNA cleavage, it completely abolished the enzyme's ability to bypass Dynabeads on bio40flapDNA. Notably, random cleavage was restored when the crosslinked Wadjet proteins were treated with the reducing agent DTT, which reopened the hinge (Fig. 2C, Supplementary Fig. 4B). Hinge closure also prevented random cleavage on all other DNA substrates tested except for obstacle-free bio40flapDNA (Supplementary Fig. 4C, D). These findings unambiguously assign the hinge domain in Wadjet-I as a gate facilitating "hinge bypass" of ssDNA and RNA-anchored obstacles. Wadjets harboring closed hinges, whether naturally occurring or artificially crosslinked, also failed to efficiently cleave DNA circles with two bio40flaps for bead attachment (Fig. 2D, Supplementary Fig. 4E). This indicates that hinge bypass is important for complete DNA extrusion and subsequent cleavage to occur, particularly in this double-obstacle scenario. Hinge closure also prevented bypass of RNA-linked obstacles (Supplementary Fig. 5A, B). Together, these findings highlight the

critical role of hinge domain in enabling loop extruding Wadjet-I to navigate around obstacles.

## Efficient obstacle bypass depends on hinge channel charges

We further characterized the SMC hinge, by first investigating whether the highly conserved positive charges that line the ecJetC hinge channel are important for hinge bypass, presumably through charge interactions with the ssDNA/RNA linker. To this end, we generated ecJET complexes with triple alanine (3A) and aspartate (3D) substitutions of ecJetC hinge channel residues R524, K591, K601 (Fig. 3B). The 3D mutant was completely defective in Dynabeads bypassing but also displayed a defect in DNA cleavage, whereas the 3A mutant exhibited a significant reduction in bypassing activity whilst retaining normal DNA cleavage efficiency (Fig. 3B). The 3A mutant further hampered the already inefficient bypassing of Dynabeads linked to the DNA circle via the neutral ethylene glycol (TEG) linker (Supplementary Fig. 6A). These results together strongly support the notion that the channel charges facilitate hinge bypass, possibly not only by capturing nucleic acid linkers and organic linkers, but also by other means such as destabilizing the hinge interfaces. We then tested the ability of the 3A mutant to restrict a 4 kb test plasmid in vivo and found that the residual bypass activity of the 3A mutant was sufficient for restriction of this substrate (Supplementary Fig. 6B). We speculate that the positive charges play a more critical role in restricting larger, more complex natural Wadjet substrates.

SMC complexes on the *B. subtilis* chromosome are thought to bypass transcription units and other SMC complexes with high but not full efficiency[33–35]. We examined the effects of mutating the bsuSmc hinge channel. While strains with bsuSmc hinge mutations (3A, 3D) showed no significant growth defects alone (Fig. 3C), they exhibited strong synthetic phenotypes when combined with the chromosome segregation mutant Δ*parB*, which removes the Smc loader ParB and by itself causes only a mild chromosome segregation defect, indicating that segregation is strongly impaired when hinge bypass is absent. Chromatin immunoprecipitation (ChIP-seq) of a bsuSmc(R583A, R643A, R645A) strain revealed highly increased KITE subunit ScpB occupancy at origin-proximal loci (Fig. 3C, Supplementary Fig. 7A, B), particularly at operons in the immediate vicinity of *parS* loading sites (Fig. 3C, Supplementary Fig. 7B), suggesting an inability to bypass active transcription complexes arising from these operons.

## ssDNA entrapment by the SMC hinge

Building on this, we sought to probe for ssDNA capture within the SMC hinge toroid and asked whether it might be a universal mechanism shared by SMC proteins from diverse species, despite their varied biological functions. Interestingly, predictions from AlphaFold3 imply a role for the positively charged residues in stabilising hinge-ssDNA interaction (Supplementary Fig. 6C). We purified isolated hinge domains from *B. subtilis* Smc (bsuSmc), *S. cerevisiae* Smc5/6 (scSmc5/6), and condensin (scSmc2/4) as well as ecJET[10,36]. Notably, the asymmetric scSmc2/4 and scSmc5/6 hinge have distinct "North" and "South" interfaces, while in the homodimeric bacterial hinges, the two interfaces have identical sequences (Fig. 3A). We incubated the purified hinge domains with either plasmid DNA or phage-derived 2.3 kb ssDNA circles, followed by crosslinking using bismaleimidoethane (BMOE), and analyzed the reaction products by protein-denaturing agarose gel electrophoresis (Fig. 4A). We found that crosslinked hinges of bsuSmc and scSmc5/6 stably entrapped ssDNA but not dsDNA circles even after protein denaturation (Fig. 4A, B, Supplementary Fig. 8A, B). ssDNA entrapment by crosslinked scSmc5/6 hinges was eliminated when the DNA was linearized prior to binding to scSmc5/6 hinges, underscoring the topological nature of this interaction (Supplementary Fig. 8C). Crosslinked hinges of scSmc2/4 as well as of ecJetC also supported entrapment of circular ssDNA in this assay (Fig. 4C, Supplementary Fig. 8D, E). Moreover, we observed that mutating

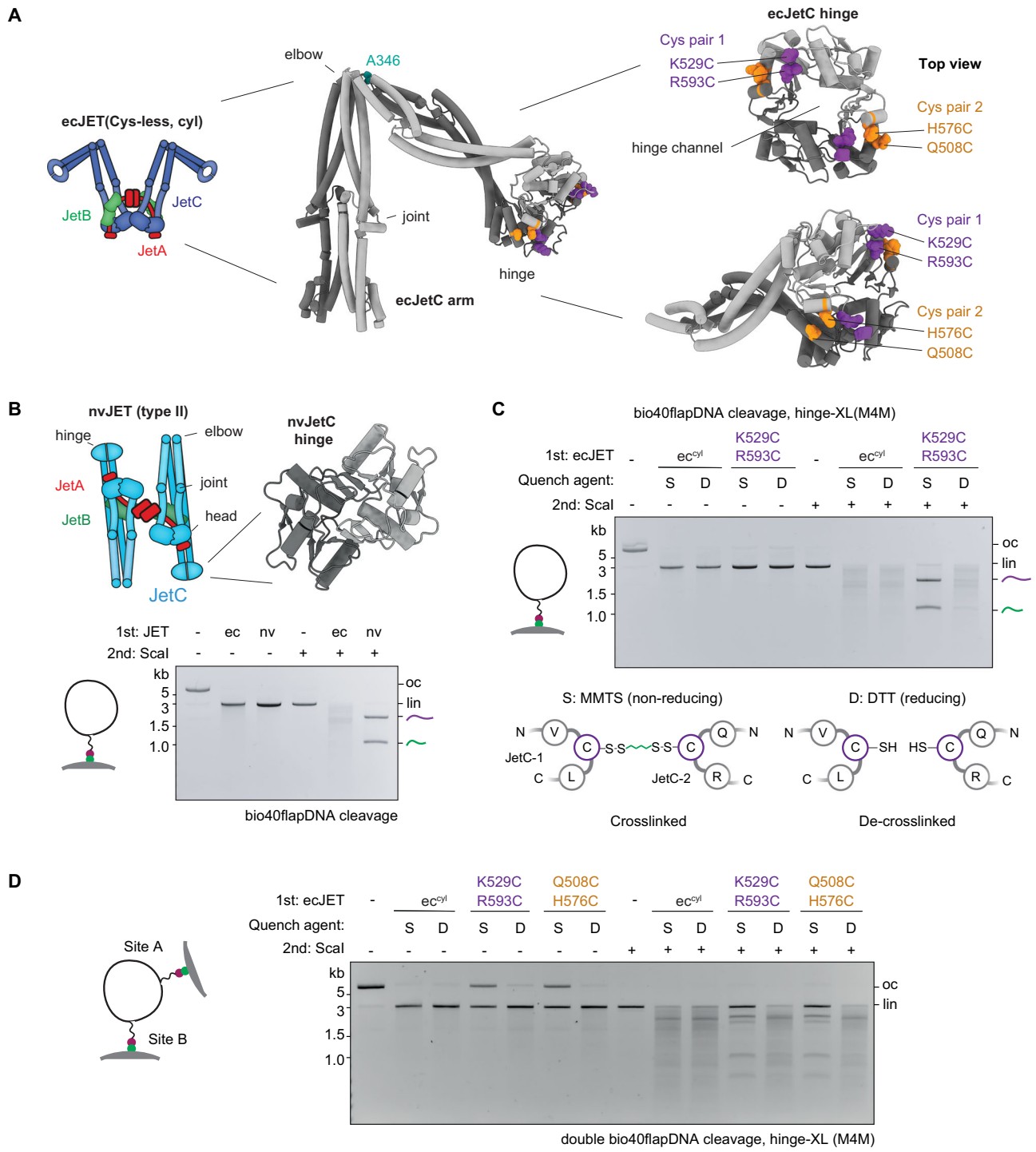

**Fig. 2 | The Wadjet-I hinge domain facilitates obstacle bypassing. A** Left: Structural model of the ecJetC arm, with the A346 residue at the elbow used for crosslinking highlighted in teal colors[30]. Right: zoomed in views of the ecJetC hinge, with the two cysteine pairs for crosslinking highlighted (pair 1: K529, R593 in purple colors; pair 2: Q508, H576 in orange colors). The hinge channel is recognizable in top view. **B** Architecture of Wadjet-II nvJET (from *N. vireti* strain LMG 21834) and a close-up model of its channel-less hinge[30]. Bottom: Agarose gel depicting the cleavage/bypass activity on bio40flapDNA attached onto Dynabeads. **C** Top: Agarose gel depicting the cleavage/bypass activity on bio40flapDNA attached onto Dynabeads by the indicated ecJET variants, after M4M-induced cysteine cross-linking and subsequent quenching by the non-reducing *S*-Methyl methanethio-sulfonate (MMTS) quencher (preserving the crosslinks) or reducing dithiothreitol

(DTT) (disrupting the crosslinks), depicted in the bottom panel (see Methods for details). Cysteine-less ecJET (ec[cyl])[31]. See Supplementary Fig. 4A for a measure of the degree of crosslinking. See panel **D** and Supplementary Fig. 4B for experiments with Cys-less ecJET harboring the other hinge crosslinking cysteine pair (pair 2). **D** Agarose gel showing the cleavage/bypass activity of the indicated ecJET variants on DNA circles with two bio40flap-Dynabeads modifications. Residual cleavage events were likely due to a population of DNA circles that have only attached onto a streptavidin-Dynabeads bead at one of the two sites. See Supplementary Fig. 4E for the experiment with Wadjet containing naturally closed hinges. oc - open circular DNA; lin - linearized DNA. Representative gel images from three technical replicates are shown. Schematics for panels **B-D** adapted from[25].

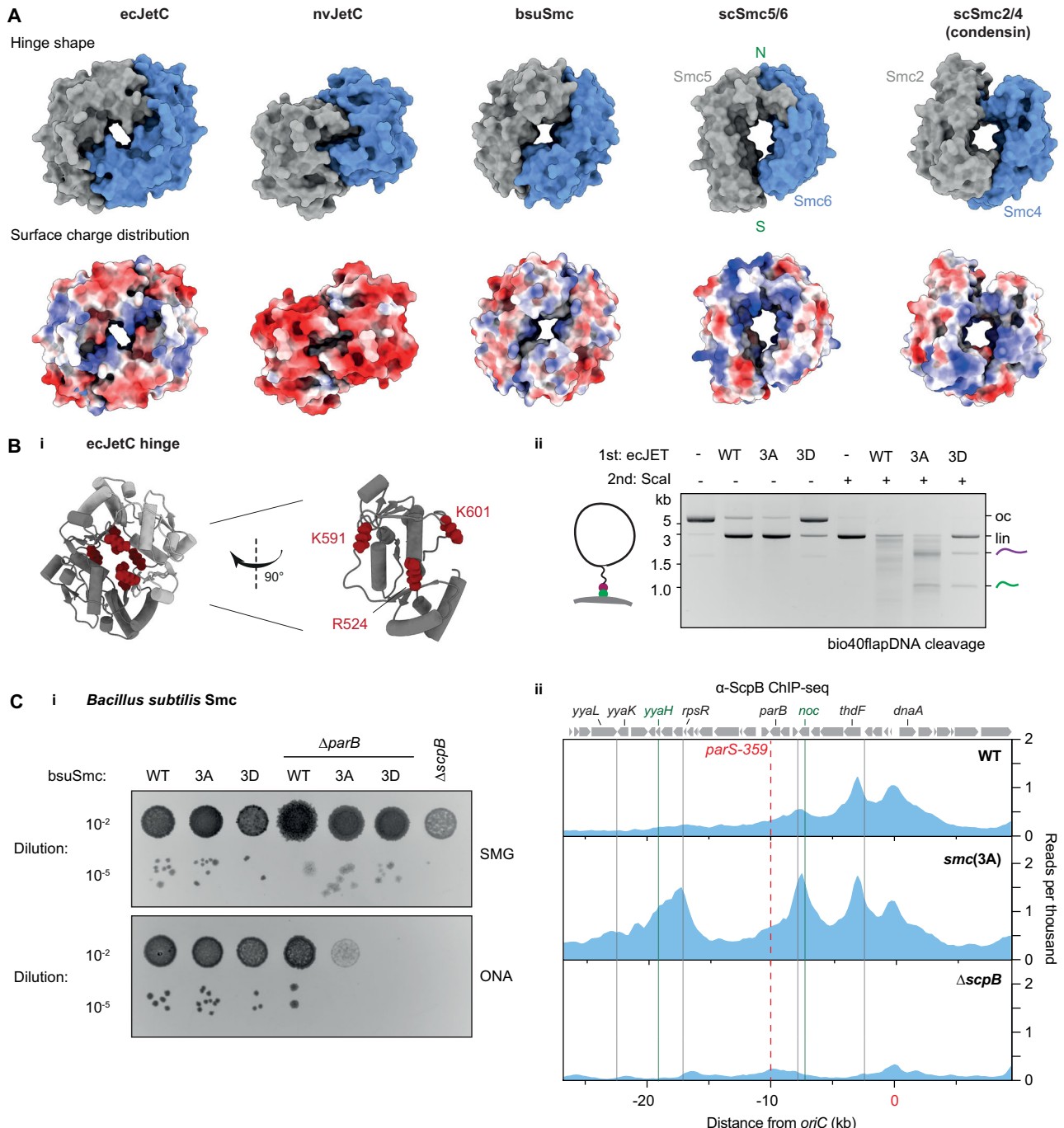

**Fig. 3 | Conservation and function of the hinge channel. A** Structural comparison of selected SMC hinges. Top: Surface view of SMC hinges used in this study. AlphaFold predictions of ecJetC, nvJetC, and bsuSmc. scSmc5/6, PDB: 7QCD[61]; scSmc2/4, PDB: 6YVU[62]. Note the absence of a notable central channel in the nvJET hinge. The "North/N" and "South/S" interfaces used in subsequent crosslinking experiments are marked for the scSmc5/6 hinge. Bottom: Surface charge distribution of the same hinges. Red colors indicating negative charges; blue colors indicating more positive charges. See Supplementary Fig. 3 for depictions of other notable SMC hinges. **B** Hinge channel charge mutants of ecJET. i) Left: Structural model of the ecJetC hinge (one domain in dark, the other in light grey colors) depicting in red colors the positively charged channel residues used for mutagenesis (R524, K591, K601). Right: An isolated ecJetC hinge domain (dark grey protomer). ii) Agarose gel depicting the cleavage/bypass activity on bio40flapDNA by the indicated ecJET variants. 3A: ecJetC(R524A, K591A, K601A); 3D: ecJetC(R524D, K591D, K601D). See Supplementary Fig. 6B for the in vivo plasmid restriction phenotypes of these mutants. **C** Hinge channel charge mutants of bsuSmc. i) Colony formation assay on minimal (SMG) or rich (ONA) media of *B. subtilis* Smc hinge mutant strains. 3A: bsuSmc(R583A, R643A, R645A), 3D: bsuSmc(R583D, R643D, R645D). Failure to grow on rich medium indicates a defect in chromosome segregation and a lethal accumulation of interlinked sister chromosomes due to insufficient SMC activity[63]. A representative agarose plate image from three technical replicates is shown. ii) α-ScpB ChIP-seq profiles for the replication origin region represented as normalized reads per thousand total reads. Gray lines indicate the 3' terminators of highly transcribed operons. Notable origin-proximal genes are also highlighted. The position of *parS-359* (a bsuSmc loading site) is given by a dashed red line. Green lines indicate the primer pair loci used for qPCR-based quantification in Supplementary Fig. 7A. See Supplementary Fig. 7B for corresponding maps for the entire chromosome. Representative examples of two biological replicates are shown.

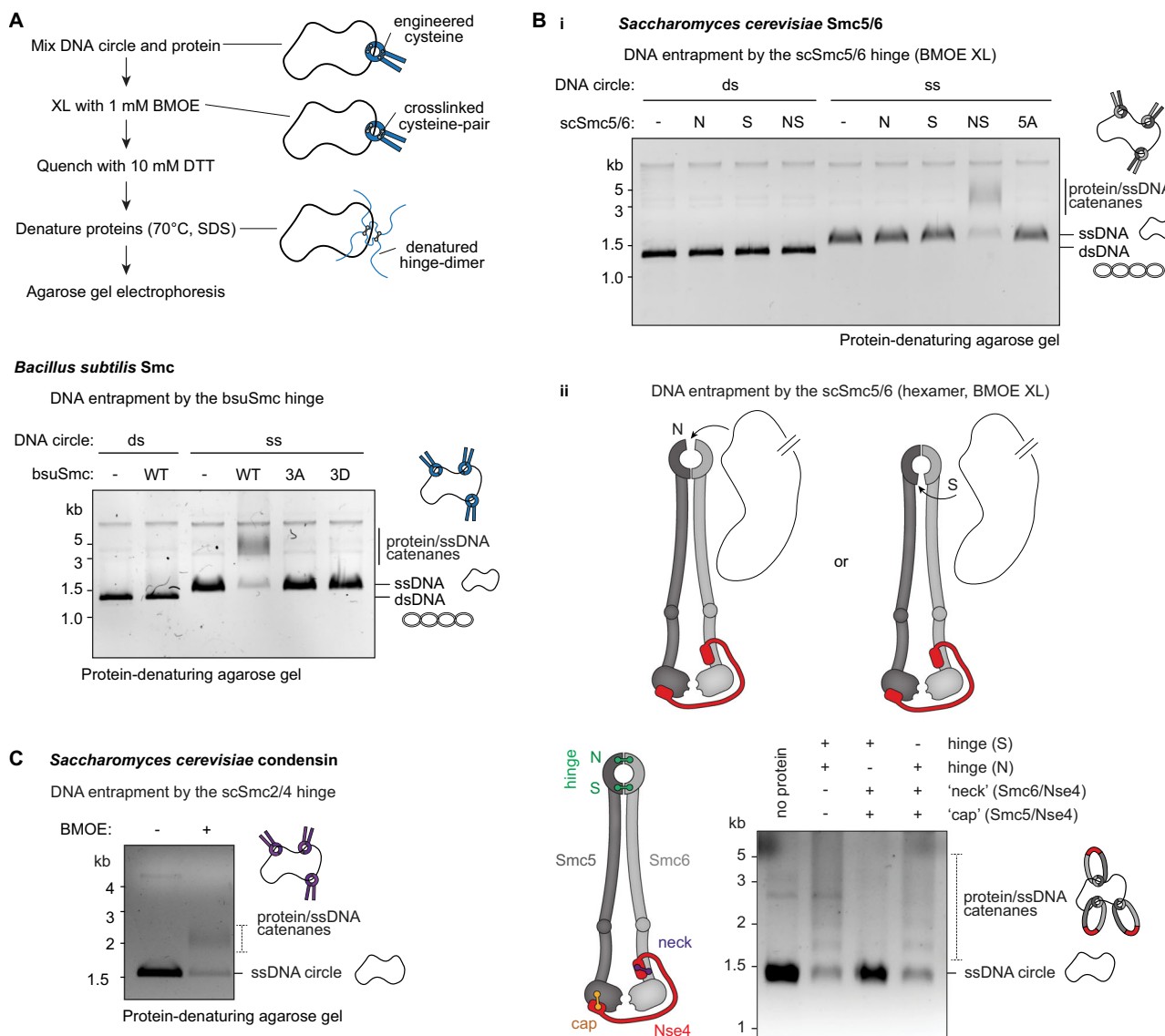

**Fig. 4 | ssDNA entrapment by selected SMC hinges. A** Denaturing agarose gel depicting 2.3 kb circular ss/dsDNA entrapment by the indicated bsuSmc hinge variants (residues 400-776). All hinges harbor C437S to mitigate non-specific crosslinking as well as engineered cysteines R558C and N634C for crosslinking by BMOE. 3A/3D contain additional alanine or aspartate substitutions (respectively) at R583, R643, R645 positions of the hinge channel. **B** ssDNA entrapment by the scSmc5/6 hinge, i) using isolated scSmc5/6 hinges and ii) scSmc5/6 hexamers. i) Denaturing agarose gel depicting 2.3 kb circular ss/dsDNA entrapment by the indicated Smc5/6 hinges (Smc5 residues 305-805, Smc6 residues 407-808), cross-linked by BMOE. N: "North" interface containing Smc5(V638C), Smc6(N572C); S: "South" interface containing Smc5(N526C), Smc6(N643C). NS: Mutant containing N and S cysteine mutations. 5A: NS mutant containing hinge channel mutations

Smc5(R537A, K612A, K641A), Smc6(R555A, R676A). ii) Left: Schematic of the tri-partite Smc5/6 ring subunits with the crosslinked interfaces marked[10]. Right: Denaturing agarose gel depicting 2.3 kb circular ssDNA entrapment by the scSmc5/6 hexamer after crosslinking the hinge (N/S interface) in combinations with crosslinking the neck and cap interfaces. See Supplementary Fig. 9 for a detailed explanation of the likely ssDNA trajectory into the Smc5/6 ring inferred from this data. **C** Agarose gel showing 2.3 kb circular ssDNA entrapment by the condensin scSmc2/4 hinge (Smc2 residues 443-740, Smc4 residues 598-923), containing cysteine residues Smc2(S560C, K639C), Smc4(V721C, M821C) for closure by BMOE crosslinking. Representative gel images from at least three technical replicates are shown. Schematics for panel **B** adapted from[10].

positively charged residues in the hinge channel of both bsuSmc and scSmc5/6 hinges abrogated ssDNA entrapment (Fig. 4A, B). These findings are consistent with the idea that capture of ssDNA inside the hinge toroid is a conserved property of all SMC complexes and that the positively charged residues lining the channel are essential for this.

We next asked how ssDNA enters the hinge toroid, and if ssDNA preferentially traverses through one of the two hinge interfaces. This is difficult to test in SMC complexes containing symmetric hinges. We therefore exploited the asymmetry of the scSmc5/6 hinge by performing ssDNA entrapment experiments in the context of the

hexameric core complex[10]. We created chemically closed compartments by individually pairing a hinge cysteine residue (North or South interface) with two cysteine pairs at the kleisin/SMC interface. Strikingly, we found that only the North combination led to efficient ssDNA entrapment (Fig. 4B, Supplementary Fig. 8F), suggesting that ssDNA likely uses the North interface as a primary gate for entering the hinge lumen (Supplementary Fig. 9). Whether this is a common feature in all asymmetric hinge containing SMCs and whether ssDNA linked with obstacles follow the same trajectory through the hinge remains to be established.

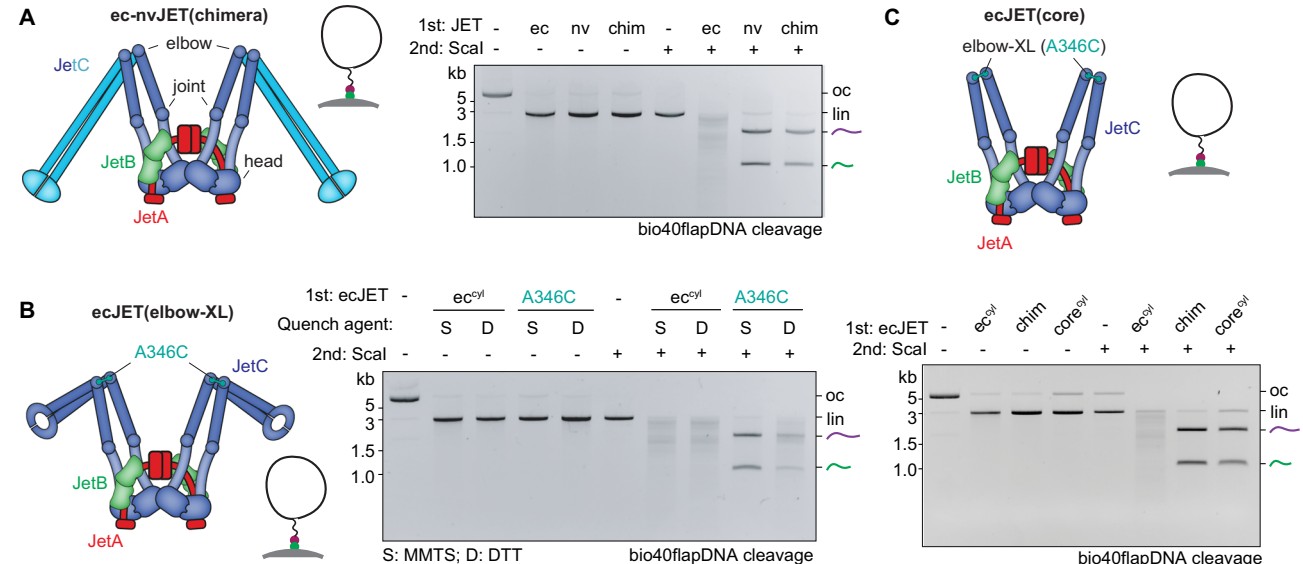

**Fig. 5 | The SMC periphery is required for obstacle bypassing but not DNA loop extrusion. A** Dynabeads bypass experiment on bio40flapDNA using a ec-nvJET chimera ('chim') harboring the ecJET core and an nvJET elbow-to-hinge segment. **B** As with A but crosslinking the ecJET elbow position (A346C) in a Cys-less ecJET background (ec^cyl). **C** As with A but with a Cys-less ecJET(core) complex harbouring A346C and lacking the elbow-to-hinge segment ('core'). The elbow-to-hinge segment is replaced by a nine-residue linker peptide (not shown). Representative gel images from three technical replicates are shown. Schematics for panels **A-C** adapted from[25].

## The SMC periphery is required for obstacle bypass but dispensable for loop extrusion

We have now established a critical role of the SMC hinge in facilitating obstacle bypass during extrusion by ecJET. However, an important question remains: does the hinge domain also play a role in extruding obstacle-free DNA? To test this, we engineered three variants of ecJET, focusing on the JetC elbow, a folding point in the middle of the SMC coiled coil[25,37]. First, we wondered whether substitution of the ecJetC elbow-to-hinge segment for the corresponding (and much larger) nvJetC segment would abolish LE and obstacle bypass. Such a chimera ('chim') was readily expressed and purified (Supplementary Fig. 10A). It retained efficient DNA cleavage (Fig. 5A), suggesting that LE activity remained intact despite the lack of a cognate elbow-hinge segment. However, it failed to bypass Dynabeads on bio40flapDNA, presumably due to the presence of the bypass-incompetent nvJET hinge (Fig. 5A). Next, we investigated whether the formation of an open elbow-to-hinge lumen is required for LE and obstacle bypass by introducing a crosslink at the JetC elbow (A346C) in Cys-less ecJET. Crosslinking at this site also blocked obstacle bypass (Fig. 5B) but had minimal impact on DNA cleavage. These findings reinforce the notion that the hinge and the hinge proximal coiled coil must remain openable for obstacle bypass. Crucially, they suggest that neither the hinge nor the associated coiled coil is necessary for LE.

To definitively rule out a requirement of the elbow-to-hinge segment for LE, we created a Cys-less ecJET core-only complex ('core') replacing the entire ecJetC elbow-to-hinge segment (aa 348-686) with a nine-residue linker peptide (GGGGSGGGG). To ensure stable dimerization of ecJetC(core) in the absence of a hinge, we included the elbow crosslink cysteine (A346C) to allow disulfide bond formation when purifying it under non-reducing conditions (Supplementary Fig. 10B). Remarkably, ecJET(core) retained the ability to cleave plasmid DNA but was completely unable to bypass obstacles (Fig. 5C). Pre-treatment with DTT eliminated DNA cleavage in this scenario (rather than rescuing bypass), indicating that the SMC ring must be closed for LE to occur (Fig. 5 C, Supplementary Fig. 10C). Collectively, these findings suggest that SMC complexes consist of a core module capable of extruding naked DNA and peripheral region (the elbow-to-hinge segment; 'SMC periphery') that independently facilitates navigation on complex DNA substrates by enabling obstacle bypass (Fig. 6A, B). This modular organization of SMC ring function, with distinct regions dedicated to different tasks, must be integrated into any comprehensive model of LE.

## Discussion

Loop-extruding SMC complexes encounter various obstacles on their chromosomal translocation tracks. Our findings define two distinct pathways for overcoming roadblocks, (i) obstacle extrusion by threading through the SMC lumen and (ii) specific obstacle bypass via the hinge gate (Fig. 6B). Smaller obstacles are putatively readily overcome by threading through the SMC lumen, such as transcription factors and nucleosomes, while others are too large for threading, exceeding the lumen size, including transcription-translation complexes in bacteria and transcription-splicing complexes in eukaryotes. The identification of the hinge as bypass gate elucidates how SMC motors navigate these larger obstacles, effectively removing size limitations provided that the obstacles contain linkages that can be accommodated by the hinge (Fig. 6A, B). Because the hinge gate allows passage only of a single nucleic acid strand at a time, the SMC ring remains sealed for dsDNA, even on obstacle-laden DNA.

### The hinge bypass gate

We define functionally separable modules within SMC complexes: a core complex responsible for DNA loop extrusion and an SMC periphery comprising the hinge and adjacent coiled coils for obstacle bypass (Fig. 6A). Our findings link the characteristic donut-shaped structure of the hinge[5] and its conservation to its role in selectively accommodating stretches of single-stranded nucleic acids and in passing them from one side of the ring to the other through a dual-gated mechanism (Fig. 6A-B). This provides a critical advantage for SMC complexes manoeuvring the crowded chromosomal environments, allowing them to get across challenging genomic regions effectively. We suspect that in cells, transcription-based ribonucleo-protein complexes—particularly at highly transcribed genes—serve as the primary substrates for hinge bypass (Fig. 6B).

The finding that both ssDNA and RNA linkages are efficient substrates for hinge bypass by ecJET, together with the observation that

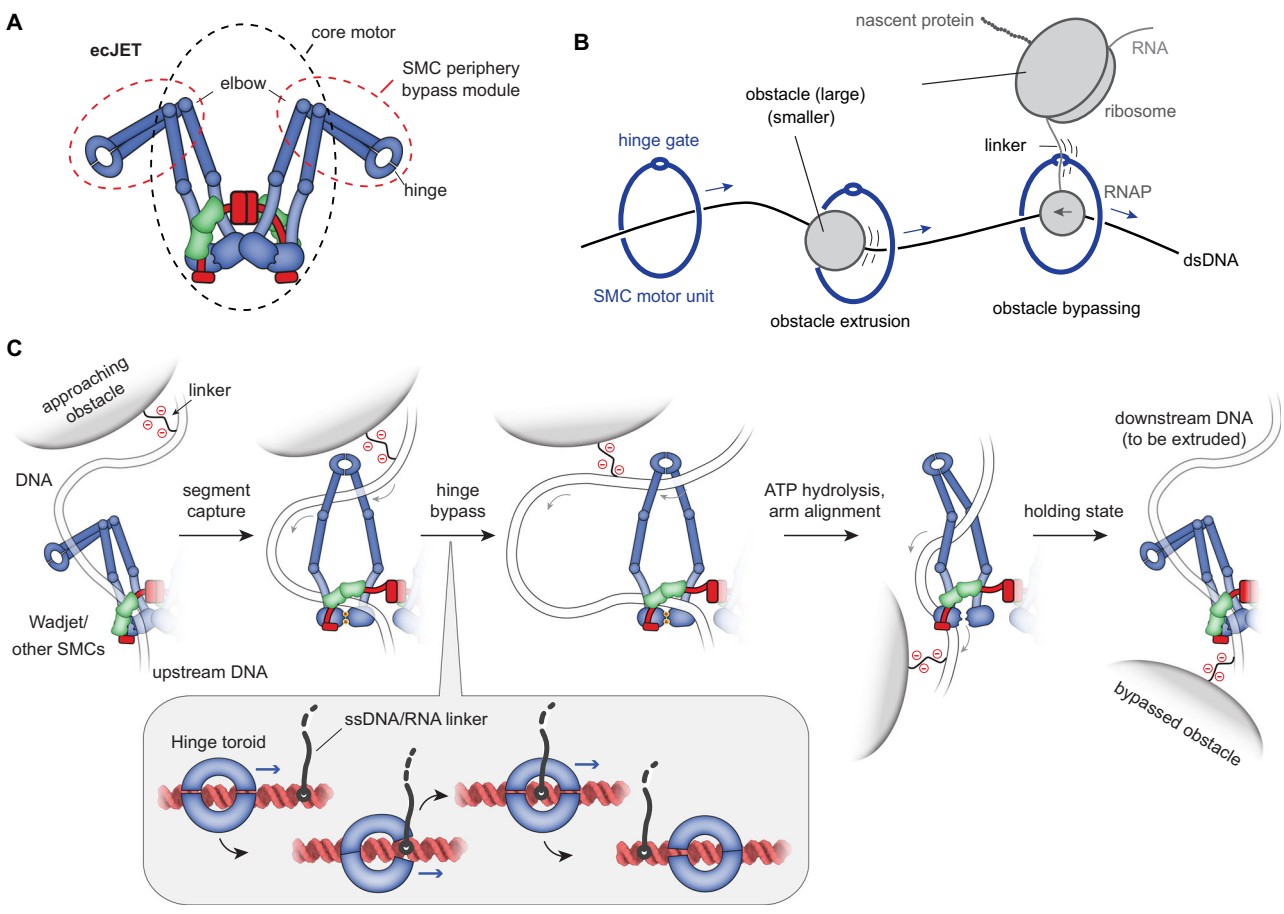

**Fig. 6 | A model for obstacle bypass by DNA loop-extruding SMC complexes.**
**A** The core and periphery of the SMC complex are responsible for loop extrusion
(LE) and for obstacle bypass, respectively. **B** Two pathways for obstacle bypass:
threading and hinge gating. **C** The segment capture+ model builds on the DNA
segment capture mechanism, which includes two alternating states: the DNA
holding state—with ATP-disengaged heads, closed coiled coil lumen and DNA held
by the kleisin compartment—and the segment capture state—with ATP-engaged

heads, open coiled coil lumen and a segment of DNA held by the SMC compartment
and clamped at the heads by the kleisin subunit. ATP hydrolysis converts the seg-
ment capture state into the DNA holding state. The segment capture+ model
incorporates an additional step for obstacle bypass during segment capture. The
stepwise passage of the obstacle linker through the double-gated hinge is illu-
strated in the close-up view. For simplicity, only a single motor unit is shown.

linkages in the form of hydrocarbon chains (Fig. 1B, Supplementary
Fig. 1B, 4C), still permit bypassing, albeit inefficiently, suggest that SMC
hinges exhibit a degree of promiscuity. The SMC-SMC bypassing
proposed for condensin based on single-molecule imaging[26] and
inferred for bsuSmc from its chromosomal dynamics[33,35] could be
explained if the respective hinge gates also accommodate peptides
that mimic ssDNA/RNA. Peptides suitable for hinge bypass may be
present in the extended unstructured regions of kleisin subunits,
particularly in cohesin and condensin. Notably, a motor unit from one
LE complex bypassing a motor unit from a neighboring LE complex
could explain the formation of interlocking 'Z-loops'[26,33,35] and how
condensin may bypass cohesive cohesin[38]. Supporting the concept of
hinge-mediated SMC-SMC bypass, hinge-substituted bsuSmc proteins
exhibit defects in chromosome folding. These defects can be sup-
pressed by reducing the number of *parS* loading sites on the chro-
mosome to one, thereby minimizing the likelihood of SMC-SMC
collisions[33].

## Two pathways for obstacle bypassing and implications for SMC evolution
Not all SMC complexes employ hinge bypass. For example, the
Wadjet II system nvJET, which has lost the central channel in its hinge,

is unable to bypass a ssDNA-linked obstacle in our assay (Fig. 2B).
Notably, like Wadjet-II, both MukBEF (Supplementary Fig. 3) and
Wadjet-III also lack a hinge channel. Strikingly, these SMC proteins—
except for a few cases within Wadjet-III—tend to possess significantly
longer coiled coils. Another example of channel-less hinges are found
in Smc proteins from *Acetobacter* and related genera (Supplementary
Fig. 3). These proteins exhibit unusually long coiled coils with the
amino-terminal coiled coil α-helix comprising 511 and 512 amino
acids in *Acetobacter pasteurianus* and *Gluconacetobacter diazo-
trophicus*, respectively, compared to 330 amino acids in bsuSmc[39].
The substantial enlargement of the SMC periphery (Fig. 6B) likely
enables these complexes to bypass larger obstacles by threading
them directly through an expanded SMC lumen, thereby at least
partially compensating for the loss of hinge-based bypass. We spec-
ulate that anti-Wadjet-I counter-defence mechanisms may have spe-
cifically targeted the hinge channel. This could have led to the
emergence of channel-free Wadjet-II and -III systems in such envir-
onments, with the lack of a hinge channel then leading to the
extension of the coiled coils to facilitate successful threading of
larger obstacles. Alternatively, a coiled coil extension, whether
occurring by chance or due to unknown selective pressures, may
have reduced the evolutionary need to maintain hinge-based bypass,

ultimately leading to the loss of the hinge donut in longer-arm Wadjet-II and -III SMC complexes.

In *B. subtilis*, the two pathways of bsuSmc obstacle bypass function alongside an obstacle avoidance mechanism mediated by the ParB protein, which loads bsuSmc complexes onto the chromosome at *parS* sites near the replication origin. bsuSmc loading at *parS* sites aligns their translocation with replication forks and most transcription units, minimizing the risk of head-on collisions[33,34]. Consequently, *parB* deletion increases reliance on hinge bypass (Fig. 3C). The presence of three distinct strategies to mitigate roadblocking underscores their physiological significance and highlights the need to investigate this process also in more complex organisms.

### Additional functions of the hinge gate in cohesin and Smc5/6

Smc5/6 has been proposed to anchor onto ssDNA to facilitate DNA repair processes, for example stabilizing ssDNA/dsDNA junctions[40,41]. In vitro, an interaction between cohesin and R loops slows down DNA translocation[42]. These observations might be explained by prolonged ssDNA/RNA entrapment by the respective hinges. The donut-shaped cohesin hinge (Supplementary Fig. 3) is also implicated in the establishment of sister chromatid cohesion in yeast[43–45] potentially involving hinge bypass of ssDNA at replication forks[46]. Supporting this, cohesin hinge channel mutants exhibit strong cohesion defects in yeast and humans[47,48]. Accordingly, the hinge bypass gate may serve as a DNA entry gate in cohesin[43].

### Outlook

By characterizing an archetypical SMC complex proficient in LE, we have identified a key molecular determinant of obstacle bypass. While the findings remain agnostic to any proposed model of DNA loop extrusion, we note that the hinge bypass integrates seamlessly into the segment capture model[16,17] (Fig. 6C). Our data demonstrate that LE proceeds uninterrupted even when the SMC arms are unable to fold at an elbow. Furthermore, we show that the hinge and the proximal coiled coils, up to the elbow, while essential for obstacle bypass, are entirely dispensable for LE itself. These findings challenge other popular models that require repeated elbow folding and unfolding of SMC coiled coils for LE and call for a reassessment of such mechanisms. Our discovery that single-stranded nucleic acids are critical for obstacle bypass opens intriguing possibilities for establishing a molecular framework for further exploration of fundamental activity in eukaryotic cells: namely how cohesin can accommodate the replication fork to generate cohesion and how condensin is able to assemble mitotic chromosomes.

## Methods

### Protein purification

**Purification of *E. coli* GF4-3 JetABCD (ecJET).** ecJetABC (WT, chimera, variants containing hinge charge-reversal mutations and cysteine-less variants) were purified similarly as previously described[25]. In brief, ecJetABC (with a 10His-TwinStrep-3C tag on JetA aminoterminal) and JetD (with a 10His-TwinStrep-3C tag on JetD carboxyterminal) were expressed separately in *E. coli* BL21 cells grown to OD 0.5 in Terrific Broth media (protein expression induction: addition of 0.5 mM IPTG, incubation 16 °C overnight). Cell lysis was performed in lysis buffer (50 mM Tris–HCl pH 7.5, 300 mM NaCl, 5% (v/v) glycerol, 25 mM imidazole, freshly supplemented with 1 mM PMSF), the lysate was then clarified by centrifugation and applied onto a 5 mL StrepTrap XT affinity column. After 5 column volumes (CV) washes with lysis buffer, elution was performed using elution buffer (20 mM Tris–HCl pH 8, 200 mM NaCl, 50 mM biotin) and ~1 mL fractions were manually collected. The relevant fractions were pooled, then the tag was removed by a 3C protease treatment (overnight at 4 °C, typically using 250 μL of 3C at 1 mg/mL for approximately 7 mL of eluate). After sample concentration (Amicon Ultracentrifugal filter units, 50 kDa

cutoff), proteins were injected onto a Superose6 Increase 10/300 GL size-exclusion chromatography (SEC) column equilibrated with ATG buffer (10 mM Hepes–KOH pH 7.5, 150 mM KOAc, 5 mM MgCl$_2$). Peak fractions were pooled, concentrated, and flash frozen in liquid nitrogen for storage at −70 °C.

For ecJetABC WT, ec-nvJetABC chimera and hinge charge reversal mutants, the initial purification steps (cell lysis, StrepTrap XT affinity chromatography, 3C tag removal) were performed with buffers supplemented with 5 mM beta-mercaptoethanol (βME) while the final size-exclusion chromatography (SEC) was performed with ATG supplemented by 1 mM TCEP.

For ecJetABC lacking native cysteine residues (Cys-less) and its derivatives containing engineered cysteine pairs, the initial purification steps (cell lysis, StrepTrap XT affinity chromatography, 3C tag removal) were performed with buffers supplemented with 5 mM βME and the final size-exclusion chromatography (SEC) was performed with ATG buffer devoid of any reducing agents.

For JetABC variants containing the ecJetC(core) (lacking the elbow-to-hinge module, aa 348–686), cell lysis was performed in lysis buffer devoid of any reducing agents but supplemented with 1 mM PMSF. After clarification, the lysate was incubated with unpacked StrepTactin sepharose XT beads (typically 750 μL for ~40 mL of lysate) and incubated on a wheel for 3 h at 4 °C. The beads were then subjected to 3 washes with lysis buffer (one wash of 50 mL followed by 2 washes of 15 mL). Elution was performed in two rounds with 5 then 2 mL of elution buffer. The resulting eluate was then filtrated to eliminate any remaining beads, and the sample was concentrated as indicated above prior to injection onto a Superose6 Increase 10/300 GL size-exclusion chromatography (SEC) column equilibrated with ATG buffer devoid of reducing agents. The tag was kept for this construct. ecJetD was purified as in ref. 25, however, with the last SEC step performed in a column equilibrated in ATG buffer without reducing agents.

Expression vectors for cysteine-less version of ecJetABC and mutants thereof containing selected cysteine pairs were constructed by 4G cloning[49]. In brief, we initially screened for complexes with natural cysteines individually replaced with non-crosslinkable residues that maintained plasmid restriction activity in vivo. Individual substitutions were then combined until a functional JetABC sub-complex devoid of any natural cysteines was obtained, with the following mutations: ecJetC(C61L, C336H, C401R, C568A, C753S, C942S, C1006S), ecJetB(C19S), and JetA(C36A)[31]. Similarly, mutations for cysteine pair generation were introduced at selected protein positions (hinge, elbow) within JetC, guided by AlphaFold2 predictions or our experimental structural models and tested for retained function both in vivo and in vitro[23,25,30]. All protein expression vectors (listed in Supplementary Data) were produced using 4G cloning[49].

**Purification of *N. vireti* LMG 21834 JetABCD (nvJET).** nvJET (from *N. vireti* LMG 21834) was purified as described in ref. 30. In brief, nvJetABC was expressed from a co-expression vector with JetA aminoterminally tagged (10His-TwinStrep-3C) and nvJetD was expressed with a carboxy-terminal 3C-TwinStrep-10His-tag. Protein expression was performed in *E. coli* BL21 as described above but using 0.25 mM IPTG for protein expression induction. Cell lysis was performed in lysis buffer freshly supplemented with 1 mM PMSF (1 mM) and 5 mM βME. After clarification, the lysate was applied onto a 5 mL StrepTrap XT affinity column, followed by a wash with 5 CV of lysis buffer. Elution was performed as decribed above using elution buffer. Relevant fractions were pooled, the tag was removed, and proteins were concentrated as described above prior injection onto a Superose6 Increase 10/300 GL size-exclusion chromatography (SEC) column equilibrated with ATG buffer (supplemented by 1 mM TCEP). For nvJetD, the clarified lysate was applied to a 5 mL HisTrap column (Cytiva) followed by a 10 CV wash in lysis buffer. A gradient elution

(10 CV) was then performed using a lysis buffer supplemented with 300 mM imidazole. The tag was removed by an incubation with 3C protease during an overnight dialysis at 4 °C into 20 mM Tris−HCl pH 7.5, 100 mM NaCl and 5 mM βME. The protein was loaded onto a 5 mL HiTrap Q HP (Cytiva), washed with fresh dialysis buffer (5 CV) and eluted by gradient elution using 20 mM Tris pH 7.5, 1000 mM NaCl 5 mM βME. Finally, the fractions containing nvJetD were concentrated and injected onto a HiLoad Superdex200 in buffer 20 mM Tris−HCl pH 7.5, 250 mM NaCl and 1 mM TCEP. After the SEC step, the peak fractions were pooled, concentrated and frozen in liquid nitrogen.

**Reconstitution of Wadjet complexes.** Once purified, reaction-ready stocks for all Wadjet variants were reconstituted by mixing ecJetABC (250 nM final dimer, $(JetC_2JetB_2JetA_1)_2$ in the case of ecJetABC; $(JetC_2JetB_1JetA_1)_2$, in the case of nvJetABC) with JetD (500 nM final dimer, $JetD_2$) in either MM buffer (25 mM HEPES-KOH pH 7.5, 250 mM potassium glutamate, 10 mM magnesium acetate), or ATG buffer (10 mM HEPES-KOH pH 7.5, 150 mM KOAc, 5 mM $MgCl_2$). Since JetABC subunits are the limiting component of the DNA cleavage reaction, all methods described below will use "ec/nvJET dimer" to refer to a preparation of ec/nvJetABC + ec/nvJetD as detailed above.

**Purification of *S. cerevisiae* Smc5/6 hexamer.** Plasmids containing wild-type or mutant versions of the six core scSmc5/6 subunits were cloned using a recently published procedure[49]. For all constructs a carboxy-terminal 3C-TwinStrep tag was added, and the complexes were purified following a published protocol[50].

**Purification of *S. cerevisiae* Smc5/6 hinge/coiled coil constructs.** Plasmids containing wild type or mutant versions of ybbR-8His-tagged scSmc6(407-808) and untagged scSmc5(364-743) were cloned using a recently published procedure[49], transformed into chemically competent Rosetta (DE3) cells, and 1 L cultures were grown in TB-medium at 37 °C to an $OD_{600}$ of 1. The temperature was then reduced to 22 °C and protein production was induced by adding IPTG to a final concentration of 0.4 mM. Cells were incubated overnight (12-16 h), harvested by centrifugation, resuspended in lysis Buffer (50 mM Tris pH 7.5, 300 mM NaCl, 5 % glycerol, 25 mM imidazole, 5 mM beta-mercaptoethanol) and lysed by sonication. The lysate was clarified by centrifugation (40000 g for 30 minutes at 4 °C) and the supernatant was loaded on a 5 ml HisTrap column (Cytiva). After washing with 10 CV of lysis buffer, the bound material was eluted with a 10 CV gradient from lysis buffer to elution buffer (lysis buffer supplemented with 500 mM imidazole). Peak fractions were analysed by SDS-PAGE, and the fractions with superior purity were concentrated to a final volume of 1 ml, typically yielding concentrations between 15 and 25 mg/ml. 500 μl aliquots were prepared and loaded on a Superdex 200 Increase 10/300 column (Cytiva) in SEC buffer (20 mM Tris pH 7.5, 250 mM NaCl, 1 mM TCEP). Peak fractions were analyzed by SDS-PAGE, pooled, and frozen in aliquots without further purification.

**Purification of *S. cerevisiae* condensin (Smc2/4) hinge/coiled coil constructs.** Budding yeast condensin hinges (scSmc2(443-740), scSmc4(598-923)) were expressed in insect Sf9 cells following the protocol described in ref. 9. They were purified by a three strep purification protocol. Cells were lysed by dounce homogenisation and lysates cleared by ultracentrifugation and the lysate collected. Protein was pulled down via its Twin-StrepII tag by incubation with StrepTactin XT resin for 2 hours at 4 °C. Protein was eluted by incubation with 100 mM biotin for 30 mins at 4 °C. The tag was cleaved by incubation with 1 mg TEV protease for 2 hours at 4 °C. Protein was then further purified by injection into a cation exchange column and eluted over a gradient of 50 mM to 1 M NaCl. Protein was then injected into a Superdex 200 increase 10/300 GL size exclusion column and the

protein peak was pooled, concentrated, and then frozen in liquid nitrogen and stored at −70 °C.

**Purification of *B. subtilis* Smc hinge/coiled coil constructs.** A construct for the *B. subtilis* hinge/coiled coil (bsuSmc(400-776; C437S, R558C, N634C))[36]; was modified using standard mutagenesis methods to introduce hinge-channel mutations. All bsuSmc hinge/coiled coil expression plasmids were transformed into BL21(DE3) Gold cells, and 1 L cultures of were grown in TB at 37 °C to an $OD_{600}$ of 1. The temperature was then reduced to 18 °C and protein expression was induced with 0.5 mM IPTG. After overnight (12-16 h) incubation the cells were harvested by centrifugation, and proteins were purified as described for the scSmc5/6 hinge/coiled coil constructs.

**Purification of *E. coli* GF4-3 JetC hinge/coiled coil constructs.** The ecJetC hinge/coiled coil, lacking endogenous cysteine residues but containing engineered cysteine residues for crosslinking, were produced in *E. coli* BL21, expressed with an amino-terminal ybbR-8His tag. A 500 mL culture of the BL21 strain was grown in TB medium at 37 °C in a non-baffled glass flask, until the culture reached $OD_{600} = 0.5$. The culture was cooled to 16 °C and protein overexpression induced by IPTG (0.5 mM final) for 16 hours. Cells were harvested by centrifugation, resuspended in lysis buffer (50 mM Tris pH 7.5, 300 mM NaCl, 5% (v/v) glycerol, 25 mM imidazole, 5 mM beta-mercaptoethanol), followed by lysis via sonication on ice with a VS70T tip using a Bandelin SonoPuls unit, at 40% output for 12 min with pulsing (1 s on / 1 s off). After clarification by ultracentrifugation (40,000 g for 30 min), the lysate was loaded onto a HisTrap HP 5 mL column (Cytiva), washed with 6 CV of lysis buffer and eluted with lysis buffer containing an increasing imidazole gradient (up to 500 mM) for 10 CV. Fractions containing ybbR-8His-ecJetC-hinge were collected, concentrated using Amicon Ultracentrifugal filter units (30 kDa cutoff) and injected onto a Superdex 200 Increase 10/300 size-exclusion chromatography (SEC) column, equilibrated with 20 mM Tris−HCl pH 7.5, 250 mM NaCl and 1 mM TCEP. Hinge-containing fractions were pooled, concentrated to about 1 mg/mL ( ~12 μM dimer) and flash frozen in liquid nitrogen for long term storage at −70 °C.

## DNA substrate preparation
**Chemically modified DNA circles.** See Supplementary Data for a list of modified substrates and their starting plasmid/oligo(s). Modified oligos (phosphorylated/biotinylated/RNA-containing) were synthesized by biomers.net GmbH. Other oligonucleotides were synthesized by Merck (VC00021). Preparation of biotinylated 2.9 kb DNA circles was done using a previously described plasmid gap-filling method[51]. 4-5 μg of pSG6970 (pG46, containing one modification site A[52], or pSG7084 (pG46-46, containing two sites A and B) was nicked in rCutsmart buffer with Nt.BbvCI (NEB, 30 U) in 20 μL reactions for one hour at 37 °C. The reaction was quenched by addition of 3 μL Tris pH 8 (200 mM) and 3 μL EDTA pH 8 (400 mM). An excess of replacement oligo (1 μL of 100 μM stock) was added to the mix. Nicked ssDNA fragments were melted away from template by heating at 80 °C for two minutes, followed by replacement oligo annealing by slowly cooling the reaction to 20 °C (1 °C decrease every minute). After column purification and elution in 52 μL water, any DNA nicks were sealed by addition of 6 μL T4 ligase buffer and 2 μL T4 ligase (10 Weiss U), followed by incubation for one hour at 22 °C. Finally, a second round of column purification was performed.

For doubly modified substrates, the replacement reaction was performed by adding two replacement oligos (1 μL each) into the tube containing nicked pSG7084. To create DNA substrates with dsDNA branches, an excess (3 μL of 100 μM stock) of annealing oligo, fully or partially complementary to the 40 nt ssDNA flap (see Supplementary Data) of STQ15-5-40ssDNAflap replacement oligo added alongside during the annealing step.

**Preparation of ssDNA circles.** Circular ssDNA substrates for entrapment assays for bsuSmc, ecJET and scSmc5/6 were prepared as previously described[53,54] with minor modifications. A phagemid based on pBluescript SK- (pDHJS4 AS-[54];) was obtained from Addgene (#78243) and transformed into competent *E. coli* DH5α cells carrying a helper plasmid (HP4_M13[55]) (Addgene #120340). Cells were plated on LB plates containing 100 μg/ml ampicillin and 50 μg/ml kanamycin, and five individual colonies were inoculated overnight (12-16 h) in 10 ml TB medium containing both antibiotics. Cultures with cells in exponential phase were then used to inoculate 3 L of the same medium in glass flasks without baffles. 5 L flasks were used but only filled with 500 ml of medium to ensure sufficient aeration. The flasks were incubated until the next morning at 37 °C shaking at 120 rpm. Bacterial cells were pelleted at 8000 $g$ for 20 minutes at 4 °C, and the supernatant was collected and subjected to a second round of centrifugation to remove as many bacterial cells as possible. The supernatant was collected in a beaker, cooled to 4 °C, and ¼ volume of 5 x phage precipitation buffer (25 % PEG 6000, 2.5 M NaCl) was slowly added while stirring the liquid with a magnetic stir bar. Precipitation continued overnight at 4 °C, and phages were then collected by centrifugation at 12000 $g$. The obtained pellet was then resuspended in phage lysis Buffer (10 mM MOPS pH 7.5, 500 mM guanidine-HCl, 1 % Triton X-100) and heated to 80 °C in a water bath for 45 min, with gentle mixing by inversion every 5 minutes. After cooling to room temperature, the obtained solution was loaded onto a column from the NucleoBond Extra Maxi Kit (Macherey-Nagel) and DNA was washed, eluted, and precipitated according to manufacturer instructions.

For the entrapment assay with scSmc2/4, the same ssDNA circles were used. pBluescript SK- first transformed into JM109 *E. coli* cells. A colony was picked and inoculated in 1 L 2x TY media plus appropriate antibiotic and cells were grown at 37 °C 200 rpm until $OD_{600}$ = 0.05. M13K07 phage was then added, and cells were grown for a further 90 min. Kanamycin was then added to a concentration of 70 μg/ml and cultures grown overnight. Cells were pelleted and the supernatant was collected. Phages were precipitated by adding 40 g PEG 6000 and 29.2 g NaCl and incubated at 4 °C for 90 mins. Phages were pelleted by centrifugation and then resuspended in 1x TE buffer. CsCl was then added to create a solution of a density of 1.35 g/ml. The solution was then centrifuged in an SW41 rotor at 280,000 $g$ at 4 °C for 24 hours. The phage content was extracted and CsCl removed by dialysis against 50 mM HEPES pH 7.5. The ssDNA was then purified by phenol/chloroform extraction followed by dialysis against 50 mM HEPES pH 7.5.

**Linearization of ssDNA circles.** Circular ssDNA and an oligonucleotide binding across its BsaI restriction site (STP840; 5′-GGTGAGCG TGGGTCTCGCGGTATCATTGCAGCACTGG-3′) were mixed in water at concentrations of 45 nM and 1000 nM, respectively. A control was included in which the oligonucleotide was omitted. Annealing was performed by heating to 70 °C for 1 minute, followed by gradual cooling to room temperature. Both solutions were then split into two separate tubes, and BsaIHF-v2 (NEB; 40 U) was added to one while the other was left untreated (Supplementary Fig. 8C). The four resulting tubes were incubated at 37 °C for 1 hour. DNA was then purified separately from all tubes using a PCR purification kit (Qiagen), and their concentrations were measured using Nanodrop spectrophotometer (Thermo Fisher Scientific). Entrapment assays were then performed as above.

**Cleavage assay with ecJET on obstacle-free DNA**
The cleavage assay performed with M4M-crosslinked Wadjets on obstacle-free bio40flapDNA (Supplementary Fig. 4D) was conducted in 15 μL (per reaction condition) MM buffer (25 mM HEPES pH 7.5, 250 mM potassium glutamate, 10 mM magnesium acetate) supplemented with ATP (1 mM final). Typically, ~7 nM DNA was mixed with

~15 nM ecJET dimer and/or ScaI (NEB, 10 U) for 15 minutes at 37 °C (per incubation step). To remove salt that hindered DNA migration and visualization in agarose gels, the completed reactions were column purified into 15 μL water. This was followed by addition of 3 μL purple loading dye (NEB). The reactions were loaded onto an EtBr-containing 1% (w/v) agarose gel, ran at 5 V/cm for 1 hour and bands visualized with a transilluminator (UVP GelSolo).

The cleavage assay performed with ecJET(core) on pDonor (Supplementary Fig. 10C) was performed in 15 μL ATG buffer (10 mM HEPES-KOH pH 7.5, 150 mM KOAc, 5 mM $MgCl_2$) supplemented with 1 mM ATP, containing 8.5 nM pDonor and 12.5 nM ecJET dimer. 2 mM final DTT was added as indicated. Reactions were incubated for 15 minutes at 37 °C, followed by addition of 3 μL 6x SDS-containing loading dye (Thermo Fisher Scientific) and heating at 70 °C for 10 minutes. The reactions were loaded onto an EtBr-containing 1% (w/v) agarose gel, ran at 5 V/cm for 1 hour and bands visualized with a transilluminator (UVP GelSolo).

**Experiments with DNA substrates anchored to streptavidin-coated Dynabeads**
**Cleavage assay with ecJET and nvJET.** Cleavage assays on Dynabeads-bound DNA were performed as described previously[25] with minor modifications. Per reaction condition, 10 μL (containing 100 μg) Dynabeads™ MyOne™ Streptavidin C1 were equilibrated and washed with 1xBW buffer (5 mM Tris-HCl pH 7.5, 0.5 mM EDTA, 1 M NaCl) according to manufacturer's protocol. The beads were resuspended in 20 μL 2xBW buffer (10 mM Tris-HCl pH 7.5, 1 mM EDTA, 2 M NaCl) and incubated with an equal volume of biotinylated DNA (200 ng in water per condition) at 30 °C for 30 minutes with shaking. After washing with 1xBW buffer, the beads were equilibrated with MM reaction buffer (25 mM HEPES pH 7.5, 250 mM potassium glutamate, 10 mM magnesium acetate).

Equilibrated beads were then resuspended in MM buffer supplemented with 1 mM ATP (15 μL per reaction, scaled accordingly). For experiments involving nvJET, 5 mM final $MnCl_2$ was supplemented to the reaction buffer. For experiments involving multiple conditions, this was split into 15 μL aliquots. Beads were treated with Wadjet (12.5 nM dimer for ecJET either freshly reconstituted or crosslinked with M4M as described below, 25 nM dimer for nvJET) and/or ScaI-HF (NEB, 10 U), 15 minutes at 37 °C per incubation step. DNA bound to the beads was eluted by addition of 85 μL preheated MM buffer supplemented with 25 mM biotin and SDS (0.1% final) and incubation for 10 minutes at 70 °C. The supernatant containing the eluate was separated from the beads with a magnetic rack. To remove salt that hindered DNA migration and visualization in agarose gels, the eluted DNA was column purified into 15 μL water. This was followed by addition of 3 μL purple loading dye (NEB). The reactions were loaded onto an EtBr-containing 1% (w/v) agarose gel, ran at 5 V/cm for 1 hour and bands visualized with a transilluminator (UVP GelSolo).

For the cleavage assay involving RNase pre-treatment (Supplementary Fig. 5B), DNA was bound to the beads as above, after splitting the beads to 15 μL reactions in MM buffer supplemented with 1 mM ATP, 1 μL of 10 mg/mL RNase A (type III-A) from bovine pancreas (Sigma) was added and incubated for 5 minutes at 37 °C. The beads were then further treated with Wadjet and ScaI as above.

**DNA leakage assay.** To determine the stability of biotinylated DNA circles on streptavidin-coated Dynabeads (DNA leakage), DNA was bound to Dynabeads™ MyOne™ Streptavidin C1 as above, and after washing and equilibration, the beads were resuspended in 15 μL MM buffer and incubated at 37 °C for 30 minutes. The supernatant containing the released fraction was isolated. The beads containing still-bound material was eluted by the addition of 85 μL preheated MM buffer supplemented with 25 mM biotin and SDS (0.1% final) for 10 minutes at 70 °C. The supernatant containing the elution fraction

was collected from the beads. Both fractions were column purified as above and loaded onto an EtBr-containing 1% (w/v) agarose gel, ran at 5 V/cm for 1 hour and bands visualized with a transilluminator (UVP GelSolo).

### in vitro protein crosslinking

**Preparative cysteine crosslinking with 1,4-butanediyl bismethanethiosulfonate (M4M).** CysLock crosslinking was performed as described previously[31]. Briefly, purified ecJetABC (Cys-less and its derivatives) were diluted to 5 µM (dimer) in ATG buffer to a volume of 25 µL. Crosslinking proceeded by adding 1.25 µL of 1,4-butanediyl bismethanethiosulfonate (M4M) (diluted to 2 mM in DMSO, LGC Standards, 100 µM final) to the protein, and the reaction was left overnight at 4 °C. The crosslinked protein solution was spun down (21000 $g$) for 10 minutes at 4 °C to remove any aggregate, its concentration was determined, and proteins were subsequently diluted to 250 nM dimer in MM buffer supplemented with quenchers, either with S-Methyl methanethiosulfonate (MMTS, Merck, 5 mM final, preserving the crosslinks) or with dithiothreitol (DTT, BioChemica, 5 mM final, disrupting the crosslinks). The quenching took place in room temperature for 30 minutes, before transfer onto ice. ecJetD (500 nM dimer final) was added to the quenched protein to produce a reaction-ready preparation.

**Analytical crosslinking of SMC hinge/coiled coil constructs with BMOE.** Crosslinking reactions for bsuSmc, scSmc5/6 and ecJET were conducted in 10 µL crosslinking buffer (20 mM Tris-HCl, 50 mM NaCl). ecJetC hinges (0.5 µM dimer final) were mixed with/without 0.5 µL of BMOE (20 mM stock, 1 mM final) for 45 seconds before quenching by addition of 1 µL DTT (100 mM stock). The crosslinked reactions were analyzed via SDS-PAGE.

For scSmc2/4, protein was diluted to 1.4 µM final in CB buffer (50 mM NaCl, 1 mM MgCl$_2$, 50 mM HEPES pH 7.5, 1 mM TCEP and 5 % glycerol) to a final volume of 10 µl. To this, either 1 µl DMSO or 1 µl of BMOE (6.4 mM stock, 0.58 mM final) was added and mixtures incubated on ice for 6 min. 4 x LDS buffer (ThermoFisher) was then added and reactions heated at 70 °C for 10 min. Reaction products were then separated by SDS-PAGE.

### DNA entrapment assays with isolated hinge/coiled coil constructs

For the entrapment assays with bsuSmc, scSmc5/6 and ecJET hinge/coiled coils, 30 µl reactions were set up containing 30 nM ssDNA or 15 nM dsDNA circles and 500 nM of protein complex in a buffer composed of 10 mM Tris-HCl pH 7.5 and 50 mM NaCl. The mixture was incubated at RT for 5 minutes before 25 µl were transferred to a fresh 1.5 mL Eppendorf tube containing 1.25 µl of 20 mM BMOE (final concentration 1 mM). After 45 seconds of incubation at room temperature, 25 µl were transferred again to another fresh 1.5 mL Eppendorf tube containing 2.5 µl of 100 mM DTT for quenching (final DTT concentration 10 mM). 5 µl of 6x loading dye with SDS (ThermoFisher) were added and the samples were heated to 70 °C for 10 minutes to denature the proteins. Aliquots of 10 µl were then separated on 1 % agarose gels (in 0.5x TBE) containing 0.03 % SDS (Sigma-Aldrich) and ran in 0.5x TBE buffer with 0.03 % SDS at 7.5 V/cm for about 1 hour at room temperature.

For the entrapment assay with scSmc2/4 hinge/coiled coils, protein was diluted to a concentration of 325 nM and ssDNA to 9.3 nM in EB buffer (20 mM NaCl, 1 mM MgCl$_2$, 50 mM HEPES pH 7.5, 1 mM TCEP and 5 % glycerol) in a volume of 13 µl. Samples were incubated at 24 °C for 60 min. To these, either 1 µl DMSO or 1 µl of 6.4 mM BMOE was added and mixtures incubated on ice for 6 min. 6x DNA loading dye (NEB) was added and mixtures heated at 70 °C for 20 min. Reaction products were then separated by electrophoresis with 0.8 % agarose gels ran at 100 V for 4 hours in 1x TBE.

### DNA entrapment assays with hexameric Smc5/6 complexes

Crosslinking reactions were set up as described for hinge/coiled coil constructs, with the exception that a modified buffer (20 mM Tris-HCl pH 7.5, 150 mM NaCl, 20 % glycerol) was used due to reduced protein stability of the hexameric core complex under low salt conditions.

### Strain construction in *E. coli*

Integration plasmids (listed in Supplementary Data) containing wild-type or hinge mutant *E. coli* GF4-3 *jetABCD* under the arabinose-inducible pBAD promoter were cloned into MFDpir *E. coli* donor. Tri-parental mating between the donor and *E. coli* BW25113 recipients was performed as previously described[56], resulting in insertion of the *jet* operons via transposon mediated integration into the recipient genome near the neutral chromosomal *glmS* loci. The test plasmid pBADMycHisA (pBAD, 4.1 kb) was introduced to these strains via electroporation at 2.0 kV. See Supplementary Data for a list of bacterial strains used.

### Plasmid stability assay in *E. coli*

Plasmid stability assays were performed as described[25]. Briefly, pBAD and *jet* operon containing *E. coli* strains were grown overnight in LB media in the presence of ampicillin (100 µg/mL final), they were diluted 1000-fold in fresh LB media without antibiotics but containing arabinose (0.02% w/v final). After 6-7 hours when cells have undergone approximately ten generations of growth, they were spotted in 10-fold dilution series on either nutrient agar (NA) plates (scoring for overall cell number), or NA plates containing ampicillin (100 µg/mL final, scoring for ampicillin-resistant plasmid-containing survivors). After overnight incubation at 37 °C, colonies were counted and extrapolated based on the dilution degree to give an estimate of the total cell number and survivor count. To quantify pBAD+ cells, the percentage of ampicillin-resistant cells over the total cell number was calculated.

### Strain construction in *B. subtilis*

Transformation with recombinant DNA was used to engineer *B. subtilis* strains at the *smc* loci by allelic replacement using starvation-induced natural competence under standard conditions (1-2 h incubation in SMM medium lacking amino acid supplements) as described in detail in ref. 57. Strains were selected on SMG-agar plates under appropriate antibiotic selection at 37 °C. Genotypes were verified for selected single colony isolates by antibiotic resistance profiling, colony PCR, and Sanger sequencing as required. See Supplementary Data for a list of bacterial strains used.

### Viability assessment by dilution spotting

*B. subtilis* cultures were inoculated in SMG medium and grown for 10 h at 37 °C under constant shaking. Cultures were diluted 1:10 in series. Dilutions of $10^2$ and $10^5$ were spotted on minimal SMG agar and nutrient agar (ONA) plates and grown at 37 °C. Colony growth was documented by imaging after 16 h for NA plates and 24 h for SMG-agar plates.

### Chromatin immunoprecipitation (ChIP)

ChIP samples were prepared as described previously[58]. Briefly, cultures were grown in 200 mL SMG at 37 °C. Cells were grown to mid-exponential phase (OD$_{600}$ = 0.02−0.03) and fixed by incubation for 30 min with 20 mL buffer F (50 mM Tris-HCl pH 7.4, 100 mM NaCl, 0.5 mM EGTA pH 8.0, 1 mM EDTA pH 8.0, 10% (w/v) formaldehyde). Cells were harvested by filtration and washed in cold PBS. The OD$_{600}$ values of the samples were normalized to 2 and resuspended in TSEMS (50 mM Tris pH 7.4, 50 mM NaCl, 10 mM EDTA pH 8.0, 0.5 M sucrose and protease inhibitor cocktail (PIC) (Sigma)) supplemented with 6 mg/mL chicken egg white lysozyme (Sigma). Samples were incubated at 37 °C for 30 min under rigorous shaking. The resulting protoplasts were harvested by centrifugation, washed in 1 mL TSEMS,

resuspended in 1 ml TSEMS and split into 3 aliquots of equivalent volume before pelleting and flash freezing.

Samples were resuspended in 2 mL of buffer L (50 mM HEPES-KOH pH 7.5, 140 mM NaCl, 1 mM EDTA pH 8.0, 1% (v/v) Triton X-100, 0.1% (w/v) Na-deoxycholate, 0.1 mg/mL RNaseA and PIC (Sigma)), transferred to 5 mL round-bottom tubes and sonicated three 20 second pulses using a Bandelin Sonoplus with an MS72 tip (90% pulse and 35% power output). Suspensions were centrifuged for 10 minutes at 21000 $g$ at 4 °C. Samples were split into 200 µL input material and 800 µL IP material. 100 ul anti-ScpB antibody serum (custom made by Eurogentec) was incubated with equivalent volumes of Dynabeads Protein G suspension (Invitrogen) for 2 h at 4 °C under gentle agitation. Beads were washed in 1 mL Buffer L directly prior to use and resuspended as 50 µL aliquots. IP material was mixed with these 50 µL aliquots and incubated at 4 °C for 2 hours under rotation.

Bound material was subsequently washed by 1 mL washes with buffer L, L5 (buffer L containing 500 mM NaCl), buffer W (10 mM Tris-HCl pH 8.0, 250 mM LiCl, 0.5% (v/v) NP-40, 0.5% (w/v) Na-deoxycholate, 1 mM EDTA pH 8.0) and buffer TE (10 mM Tris-HCl pH 8.0, 1 mM EDTA pH 8.0). Beads were resuspended in 520 µL TES (50 mM Tris-HCl pH 8.0, 10 mM EDTA pH 8.0, 1% (w/v) SDS). Input material was supplemented with 300 µL TES and 20 µL 10% (w/v) SDS. Tubes were incubated at 65 °C overnight under vigorous shaking. DNA was purified using two rounds of phenol-chloroform extraction. 400 µL of extracted DNA was mixed with 1.2 µL GlycoBlue (Invitrogen), 30 µL of 3 M Na-acetate (pH 5.2) and 1 mL ethanol (filtered). Samples were incubated at −20 °C for 20 minutes. Precipitated DNA was pelleted by centrifugation at room temperature at 21000 $g$ for 10 minutes and was dissolved by incubation at 55 °C in 100 µL EB (Qiagen) under vigorous shaking for 10 minutes. Samples were subsequently purified using a PCR purification kit (Qiagen) as per protocol and eluted in 50 µL EB.

For quantification by qPCR, samples were diluted to 1/10 for IP and 1/1000 for input material. Reactions for qPCR were prepared by mixing 4 µL diluted samples with 5 µL 2 × 5 µL Takyon SYBR MasterMix and 1 µL qPCR primer mixture (3 µM). A list of qPCR primers is given in Supplementary Data. Samples were run in a Rotor-Gene Q (Qiagen) and analyzed using PCR miner[59].

For ChIP-seq, DNA libraries were prepared by Genomic Technologies Facility at CIG, UNIL, Lausanne. Briefly, the DNA was fragmented by sonication (Covaris S2). DNA libraries were prepared using the Ovation Ultralow Library Systems V2 Kit (NuGEN) with 18 cycles of PCR amplification. 20-30 million single-read sequences were obtained by an AVITI sequencer (Element Biosciences), single read run at 150 bp read length.

**ChIP-seq data analysis.** Sequencing data from ChIP and input samples were mapped and sorted (by coordinate) to the Bsu reference genome NC_000964 (centered on its first coordinate) using Bowtie2 (sensitive settings) and samtools sort respectively on the Galaxy project website (https://usegalaxy.org/)[60]. Reads were filtered for mapping quality (MAPQ) greater than 10, reduced to bins of 250 bp, and normalized for total read count in SeqMonk. The data were exported and visualized in Graphpad Prism 10.

#### Reporting summary
Further information on research design is available in the Nature Portfolio Reporting Summary linked to this article.

## Data availability
All raw data (including gel images, quantification of plasmid restriction, and AlphaFold3 outputs) are available and accessible through Mendeley Data at https://doi.org/10.17632/jshbprb4sh.1 [https://data.mendeley.com/datasets/jshbprb4sh/1]. The ChIP-seq data generated in this study has been deposited in the NCBI GEO database under the accession code GSE303091. Source data are provided with

this paper. Published structures are available at the Protein Data Bank under the accession codes: scSmc5/6 hinge, PDB: 7QCD; scSmc2/4 hinge, PDB: 6YVU; mmCohesin hinge PDB: 2WD5; ptMukB hinge, PDB: 7NYY. Source data are provided with this paper.

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

## Acknowledgements

We are grateful to Stéphane Marcand and members of the Gruber Lab for critical feedback on the manuscript and for stimulating discussions. Sequencing of ChIP samples was performed at the Lausanne Genomic Technologies Facility (GTF). This study was supported by the European Research Council (724482 to S.G.), the Swiss National Science Foundation (170242 to S.G.), and by the Wellcome Trust (226494/Z/22/Z to M.S.). H.W.L. and F.R.-H. were supported by EMBO Postdoctoral fellowships (ALTF 490-2021 and ATLF 302-2022).

## Author contributions

Formal analysis: H.W.L., F.R.-H., M.T.; Investigation: H.W.L., F.R.-H., M.T., J.C.; Methodology: H.W.L., F.R.-H., M.T., J.C.; Materials: core-, chim-, cysless-ecJET: F.R.-H.; condensin hinge: J.C.; other hinges: M.T., H.W.L., F.R.-H.; DNA substrates: H.W.L., M.T.; Supervision: S.G., M.S;

Visualization: H.W.L., F.R.-H., M.T., S.G.; Writing – original draft: H.W.L., S.G.; Writing – review & editing: H.W.L., F.R.-H., M.T., J.C., M.S., S.G; Funding acquisition: S.G., H.W.L., F.R.-H., M.S.; Conceptualization: S.G.

## Competing interests

The authors declare no competing interests.
