## [Transparent Peer Review file · Nature Communications]

The SMC Hinge is a Selective Gate for Obstacle Bypass

Corresponding Author: Dr Stephan Gruber

Version 0:

Reviewer comments:

Reviewer #1

(Remarks to the Author)

The authors of the manuscript „The SMC Hinge is a Selective Gate for Obstacle Bypass“ used an elegant system to explore the function of the hinge channels in different SMC complexes. This approach enabled them to discover their new role in bypassing ssDNA/RNA-anchored complexes. The advantage of their system was that they could follow two parameters, efficiently controlling the quality of the proteins. In this way, their data are very robust and publishable with no need for additional controls/revisions.

Minor issues:

1. “RESULTS” title is missing
2. Figures should follow the text flow (Extended Data Fig 10 is cited after Extended Data Fig 2, line 133)
3. All experimental procedures must be described. For example, a description of the purification of proteins is not sufficient as it lacks details on the cloning of the recombinant plasmids.
4. What could be other reasons for the evolutionary correlation between longer SMC arms and non-channel hinges (page 10)?

Reviewer #2

(Remarks to the Author)

In the manuscript: ‘The SMC hinge is a selective gate for obstacle bypass’ by the Gruber lab, the authors describe how SMC complexes can bypass large obstacles by using their hinge domain. They focus first on the Wadjet complex, and show that only 2 nucleotides are needed to bypass an obstacle. Dynamic opening of the hinge domain is required for this bypass, as is the positive charge inside of the hinge. The authors show that this principle is not only used by prokaryotic SMC complexes, but that the hinge of yeast SMC5/6 and condensin entrap single-stranded DNA in a similar fashion. The authors end with a chimera SMC complex assembled out of two different Wadjet complexes, showing that neither the hinge nor the coiled-coils are needed for loop extrusion, but are needed for plasmid cleavage.

I think the experiments throughout this manuscript are very elegant. The manuscript reveals how the complex can extrude on DNA that is bound by many proteins. The authors hereby address an issue that was a long-standing open question for the field. I am in support of publication of this manuscript, and I have only have some minor points that should be addressed:

- 1) In figure 1 the authors introduce an assay that is used many times throughout the manuscript. However the first lane for me already raises questions. In figure 1b the first lane, two bands can be observed: 1 around 7 kb and one at 2 kb. Why does the Biotin (int dT) have this extra band at 2 kb, while Biotin (end) does not have to this band? Please explain.
- 2) Figure 1 shows that besides DNA also RNA can be used by the complex to bypass a roadblock. Could a short (acidic?) stretch of protein also follow the same trajectory? If so, this would directly lead to many more interesting scenarios by which DNA bound proteins can be bypassed.
- 3) Figure 3B i): Here it would help if there would be a small arrow indicating the turn that is made.
- 4) Figure 3E i): Here it would help to indicate in these figures which kind of gel is used, i.e. a denaturing gel. That would help explain why crosslinking is needed to see it on this gel.

- 5) Figure 3D i) and ii): These figures would benefit from a schematic depiction of the order of events in the experiment.
- 6) Figure 3E: The dotted line to show indicating protein/ssDNA catenanes is at the wrong position – Presumably should be lower.
- 7) A main feature that is pointed out and tested by the authors is the donut shape of the hinge, which is even absent in nvJET. It would help if this is explained and featured in their schematic representation of the complex throughout the manuscript. Especially in figure 4 this would help explain the purpose of the chimera.
- 8) The authors introduce the nvJET as a donut-less SMC complex. In the discussion they suggest interesting ideas why this might be, but here they don't indicate that nvJET is a Wadjet II type SMC complex. Best to make this clearer for the reader.
- 9) Lines 307/308 were unclear to me. Please rephrase.

Reviewer #3

(Remarks to the Author)

In the present work, Liu et al. describe a series of biochemical experiments aimed at understanding how SMC proteins bypass large obstacles on DNA. Using *E. coli* WadjetABCD as a model system and a bead-coupled plasmid cleavage assay that relies on loop extrusion by the JetC ATPase, Wadjet is found to be capable of most efficiently bypassing the bead linkage when a 5' ssDNA, ssRNA or a hydrocarbon linker is present. Crosslinking studies indicate that a dimerization interface present in a coiled-coil region of JetC (known as the 'hinge') needs to be capable of opening for bypass to occur. Structural comparisons show that the ecJetC hinge has an interior, positively charged channel that is present in many other SMC homologs including bacterial Smc and eukaryotic SMC1/3 (cohesin), SMC2/4 (condensin), and SMC5/6, but not in all Wadjet systems or bacterial MukBs. Analysis of a naturally 'channel-less' Wadjet (nvJET) hinge reveals that this system supports loop extrusion but not obstacle bypass; replacement of the ecJetC hinge with that of nvJetC is shown to reprogram ecJET to prevent bypass while leaving loop extrusion and DNA cleavage functions intact. Moreover, replacement of three lysines in the ecJetC hinge pore with alanine abolishes obstacle bypass but not loop extrusion and DNA cleavage, as does replacement of the hinge and a portion of the JetC coiled coil with a Gly/Ser linker.

Excepting some minor confusion with figure panel callouts (noted below), the paper is well-written and readily accessible to a general audience. The experiments are clever and successfully tackle one of the more poorly understood aspects of SMC protein function (which is saying something, given the enigmatic nature of these factors) to reveal new and significant insights into the mechanisms that underpin obstacle bypass. The conclusions drawn from the data are compelling and represent a fundamental advance for the field. Pending resolution of a few issues, publication is recommended.

Specific comments/questions

Is there any evidence that the presence of a suitable polymer (e.g., ssDNA, etc.) in the JetC coiled-coil arm lumen actively destabilizes the hinge dimer interface to more readily promote bypass?

Will the triple Lys-to-Ala mutant bypass the int-TEG linkage? What happens if a PNA linkage? It might be possible to address a few of the speculative points raised in the discussion with these relatively straightforward experiments.

Fig. 3Di and 3E. Can the formation of ssDNA-protein catenanes be confirmed by adding a short complementary oligo to the ssDNA with a unique restriction site and cutting the ring?

Minor points

Fig. 1B. Why does the no protein control for the first substrate look different from the other three? Is it because the DNA supercoiled in this instance and nicked in the others? Please clarify in the figure panel and legend.

Fig. 2A, right, lower - please label which cysteines are which.

Fig. 3Cii. What do the two rows of each spotting assay correspond to?

Line 128. It's a bit confusing to call out Fig. 3A before 2B or EDFig8 before EDFig. 3-7. Please reorder.

Line 154. EDFig. 4A-B is called out before EDFig. 3E is discussed in the text.

Legends 3C and 3D are conflated.

Version 1:

Reviewer comments:

Reviewer #1

(Remarks to the Author)

The authors of the manuscript „The SMC Hinge is a Selective Gate for Obstacle Bypass“ revised it accordingly. I found the manuscript of excellent quality and publishable.

Reviewer #3

(Remarks to the Author)

The author of satisfactory addressed all questions raised during the prior review. No further revisions are recommended.
